REGISTERED REPORT

# Registered report: The common feature of leukemia-associated IDH1 and IDH2 mutations is a neomorphic enzyme activity converting alpha-ketoglutarate to 2-hydroxyglutarate

Oliver Fiehn[1], Megan Reed Showalter[1], Christine E Schaner-Tooley[2], Reproducibility Project: Cancer Biology*

[1]West Coast Metabolomics Center, University of California, Davis, Davis, United States; [2]University of Louisville School of Medicine, Louisville, United States

REPRODUCIBILITY
PROJECT
CANCER BIOLOGY

*For correspondence: stephen@cos.io

Group author details: Reproducibility Project: Cancer Biology See page 18

**Abstract** The Reproducibility Project: Cancer Biology seeks to address growing concerns about reproducibility in scientific research by conducting replications of selected experiments from a number of high-profile papers in the field of cancer biology. The papers, which were published between 2010 and 2012, were selected on the basis of citations and Altmetric scores (*Errington et al., 2014*). This Registered Report describes the proposed replication plan of key experiments from "The common feature of leukemia-associated IDH1 and IDH2 mutations is a neomorphic enzyme activity converting alpha-ketoglutarate to 2-hydroxyglutarate" by Ward and colleagues, published in Cancer Cell in 2010 (*Ward et al., 2010*). The experiments that will be replicated are those reported in Figures 2, 3 and 5. Ward and colleagues demonstrate the mutations in isocitrate dehydrogenase 2 (*IDH2*), commonly found in acute myeloid leukemia (AML), abrogate the enzyme's wild-type activity and confer to the mutant neomorphic activity that produces the oncometabolite 2-hydroxyglutarate (2-HG) (Figures 2 and 3). They then show that elevated levels of 2-HG are correlated with mutations in *IDH1* and *IDH2* in AML patient samples (Figure 5). The Reproducibility Project: Cancer Biology is a collaboration between the Center for Open Science and Science Exchange and the results of the replications will be published by *eLife*.

## Introduction

Mutations in the metabolic enzymes isocitrate dehydrogenase 1 (*IDH1*) and *IDH2* genes, which catalyze the production of α-ketoglutarate (α-KG) from isocitrate, have been associated with numerous forms of cancer (*Krell et al., 2013*) leading to exploration of how changes in their function could be linked to the development of tumors. All known mutations alter key residues in both proteins that decrease the enzyme's affinity for isocitrate, leading to the theory that the loss of IDH function perturbs the equilibrium of α-KG, negatively affecting various α-KG dependent enzymes (*Zhao et al., 2009*). However, work from the Thompson group determined that the tumor-associated mutations actually created a neomorphic function; rather than catalyzing the production of α-KG, mutant IDH proteins produce the oncometabolite 2-hydroxyglutarate (2-HG) (*Ward et al., 2012*). Dang and colleagues first described this neomorphic function and demonstrated a correlation between 2-HG levels and glioma samples harboring *IDH* mutations (*Dang et al., 2009*). In their 2010 Cancer Cell paper, Ward and colleagues further confirm these findings and extend the association of 2-HG levels and *IDH* mutations to acute myeloid leukemia (AML) (*Ward et al., 2010*).

In Figure 2, Ward and colleagues transfected 293T cells with either wild type or mutant forms of *IDH2*. They assessed cell lysates for their ability to generate NDPH in the presence of isocitrate (Figure 2A) or to consume NADPH in the presence of α-KG (Figure 2B). Their data indicated that cells transfected with IDH2[WT] generated NADPH in the presence of isocitrate, and did not consume much NADPH in the presence of α-KG, consistent with its canonical function of converting isocitrate to α-KG. However, IDH2[R172K] displayed the opposite effect, indicating that it was able to consume NADPH in an α-KG dependent manner. These data were the first suggesting that the mutant form of IDH2 might have a neomorphic function. This key experiment will be replicated in Protocol 1.

In Figure 3, Ward and colleagues use gas-chromatography mass spectrometry (GC-MS) to identify a novel function of IDH2[R172K]. They identified a unique peak in the lysates of cells transfected with IDH2[R172K] that corresponded to the retention time of the metabolite 2-hydroxyglutarate (2-HG). They confirmed the metabolite identity by mass spectrometry. These data provide evidence that the mutant form of IDH2 leads to 2-HG production. This key experiment will be replicated in Protocol 2.

In Figure 5, Ward and colleagues examined the correlation between AML patient samples carrying *IDH* mutations and the levels of 2-HG found in those samples. They showed that patient samples carrying *IDH* mutations contained higher levels of 2-HG than samples from patients with WT *IDH* genes. This key experiment will be replicated in Protocol 3.

Several groups' work has supported the results of Ward and colleagues, who themselves confirmed and extended their initial findings in subsequent reports (*Ward et al., 2011*; *2013*). Leonardi and colleagues confirmed that mutant forms of *IDH*, specifically *IDH1*, did not perform the canonical forward reaction converting isocitrate to α-KG (*Leonardi et al., 2012*). Using magnetic resonance spectroscopy, Izquierdo-Garcia and colleagues confirmed that transfection of cells with mutant *IDH* forms increased the levels of 2-HG (*Izquierdo-Garcia et al., 2015*), while Jin and colleagues demonstrated similar findings for *IDH1* and *IDH2* mutants (*Jin et al., 2011*). Evaluating 2-HG levels in astrocytomas and gliomas harboring various *IDH1* mutations, Pusch and colleagues also showed that any mutations in *IDH1* correlated with increased levels of 2-HG in human patient samples (*Pusch et al., 2014*), a trend also observed by Juratli and colleagues (*Juratli et al., 2013*).

Discovery of IDH neomorphic function, resulting in the production of the 'oncometabolite' 2-HG, opened many avenues of research into how the production of excess 2-HG could impact tumorigenesis. Figueroa and colleagues expanded upon the foundation laid by Ward and colleagues and determined that excess 2-HG was correlated with changes in global methylation patterns (*Figueroa et al., 2010*). Xu and colleagues showed that 2-HG was able to competitively inhibit many α-KG dependent enzymes, including several histone demethylases, and that exogenous 1-HG was able to inhibit histone demethylation (*Xu et al., 2011*). Lu and colleagues also observed this correlation between 2-HG levels and perturbations in global histone methylation patterns, and went on to show that this resulted in impaired cellular differentiation (*Lu et al., 2012*).

## Materials and methods

Unless otherwise noted, all protocol information was derived from the original paper, references from the original paper, or information obtained directly from the authors.

### Protocol 1: Assessing the α-ketoglutarate dependent NADPH consumption of wild-type or mutant IDH2

In this protocol, 293T cells are transfected with empty vector, IDH2[WT], or IDH2[R172K]. Lysates are generated from these cells and their ability to produce NADPH from NADP+ and isocitrate is assayed (Figure 2A). The same lysates are also assayed for their ability to consume NADPH in the presence of 0.5 mM α-ketoglutarate (α-KG) (Figure 2B). Expression of the transfected protein will be confirmed by Western blot (Figure 2C).

### Sampling

Oxidative and reductive activity (Figures 2A and B):

- Experiment has three conditions. Each will be performed with seven biological replicates and three technical replicates of each condition at each time point for a final power of at least 80%.

- Condition 1: 293T cells expressing *IDH2*^WT
- Condition 2: 293T cells expressing *IDH2*^R172K
- Condition 3: 293T cells expressing empty pCDNA3 vector
  o Each lysate will be assessed for cell's ability to reduce $NADP^+$ and oxidate NADPH
  o See Power Calculations section for details.

Confirmatory Western Blot (Figure 2C)

- This is a quality control experiment and is not being powered to detect a specific effect size. Western blots will be performed alongside each biological replicate.
- Western blotting of each lysate will be performed for the following proteins
  - IDH2
  - IDH1
  - Actin [additional]

## Materials and reagents

| Reagent | Type | Manufacturer | Catalog # | Comments |
|---|---|---|---|---|
| 293T cells | Cells | ATCC | CRL-3216 | Original source unspecified |
| Dulbecco's modified Eagle's medium (DMEM) | Media | Invitrogen | 11965118 | Original unspecified |
| FBS | Reagent | Hyclone | SH30071.03 | Replaces FBS from CellGro |
| IDH2^WT ORF in pCMV6 | Plasmid | Origene | RC201152 | |
| IDH2^R172K ORF in pCMV6 | Plasmid | Origene | RC400103 | |
| pCDNA3 | Plasmid | Invitrogen | V790-20 | |
| Lipofectamine 2000 | Reagent | Invitrogen | 11668027 | |
| M-Per Mammalian protein extraction reagent | Reagent | Pierce | 78503 | |
| Aprotinin | Reagent | Sigma | 248614 | Original protease inhibitor cocktail unspecified |
| AEBSF | Reagent | EMD Millipore | 101500-100MG | |
| Leupeptin | Reagent | Sigma | L2884-100mg | |
| Pepstatin A | Reagent | EMD Millipore | 516481-100MG | |
| NaOV | Reagent | Sigma | 450243-50G | Original unspecified |
| NaF | Reagent | Sigma | 215309-50G | |
| Sonicator | Equipment | VCR | 75HT | Original unspecified |
| Refrigerated microcentrifuge | Equipment | Labnet International, Inc | PrismR | Original unspecified |
| Tris-HCl | Reagent | BioRad | BR0011 | Original unspecified |
| MnCl2 | Reagent | M87-100 | Fisher | Original unspecified |
| EDTA | Reagent | VWR | EM-4050 | Original unspecified |
| ß-NADP+ | Reagent | MP Biomedicals | ICN10116680 | Original unspecified |
| ß-NADPH | Reagent | Sigma | 10107824001 | Original unspecified |
| D-(+)-threo-isocitrate | Reagent | Sigma | I1252 | |
| Spectrophotometer | Instrument | Molecular Devices | Filter Max F5 Multi-mode Microplate Reader | Original unspecified |
| 6-well tissue culture plates | Materials | E& K Scientific | 27160 | Original unspecified |
| 96 well plates | Materials | Fisher (Costar) | 07-200-656 | Original unspecified |
| Tric-HCl | Reagent | BioRad | BR0011 | Original unspecified |

*Continued on next page*

*Continued*

| Reagent | Type | Manufacturer | Catalog # | Comments |
|---|---|---|---|---|
| Glycerol | Reagent | VWR | EM-4760 | Original unspecified |
| ß-mercaptoethanol | Reagent | Sigma | M6250-250mL | Original unspecified |
| Sodium dodecyl sulfate (SDS) | Reagent | Sigma | L3771-100G | Original unspecified |
| Bromophenol blue | Reagent | Sigma | B0126-25G | Original unspecified |
| Protogel | Reagent | Fisher/National Diagnostics | 50-899-90119 | Original unspecified |
| APS | Reagent | Sigma | 248614 | Original unspecified |
| TEMED | Reagent | Fisher | BP150-100 | Original unspecified |
| nitrocellulose | Materials | BioRad | 162-0112 | Original unspecified |
| Anti-IDH2 antibody (mouse monoclonal) | Primary Antibody | Abcam | ab55271 | |
| Anti-IDH1 antibody (goat polyclonal) | Primary Antibody | Santa Cruz | sc49996 | |
| Anti-Actin antibody (rabbit monoclonal)-HRP conjugated | Primary Antibody | Cell Signaling | 12620 | Not included in original. |
| ECL Mouse IgG, HRP-linked whole Ab (from sheep) | Secondary Antibody | GE Healthcare | NA931V | |
| HRP conjugated rabbit anti-goat antibody | Secondary Antibody | Invitrogen | 811620 | Original unspecified |
| Protein ladders | Reagent | Cell Signaling Tech. | 7727L | Original unspecified |
| | | Gold Biotech | p007-1500 | Original unspecified |
| ECL reagent | Reagent | Fisher Scientific | PI34096 | Original unspecified |
| Endo-free maxiprep kit | Reagent | Qiagen | 12362 | Original unspecified |
| α-ketoglutarate | Reagent | Sigma | 75892-25G | Original unspecified |
| DC Protein Assay Kit | Kit | BioRad | 5000112 | Original unspecified |
| Alpha innotech imager | Equipment | Alpha Innotech | Alphaimager 2200 | |
| sodium azide | Reagent | Sigma | S2002-5G | Original Unspecified |
| Ponceau stain | Reagent | Quality Biological | 50-751-6798 | Additional reagent |

## Procedure

### Notes

- 293T cells are grown in DMEM supplemented with 10% FBS at 37°C in 5% $CO_2$
- Cells will be sent for STR profiling and mycoplasma testing.

1. Confirm insert identity by sequencing.
   a. Origene clones are shipped with two sequencing primers.
2. Sub-clone $IDH2^{WT}$ and $IDH2^{R172K}$ from the Origene pCMV6-Entry vectors into pcDNA3.
   a. Confirm insert identity by sequencing.
   b. Confirm vector integrity by agarose gel electrophoresis.
3. Grow up and use an endo-free maxiprep kit to prep the following vectors:
   a. pcDNA3
   b. pcDNA3-$IDH2^{WT}$
   c. pcDNA3-$IDH2^{R172K}$
4. Seed $0.25-1x10^6$ 293T cells per well of a 6-well plate in growth medium without antibiotics.
   a. Grow overnight.
   b. Confirm cells at 70–80% confluency by light microscopy at time of transfection.
5. Transfect 293T cells with pcDNA3, pcDNA3-$IDH2^{WT}$, pcDNA3-$IDH2^{R172K}$ with Lipofectamine 2000 according to manufacturer instructions for a 6-well plate.

 a. As per manufacture's instructions 1 µg plasmid DNA per well in a 6-well plate for 70–80% confluent 293T cells.

 b. Transfect 1 well (or plate if reaction needs to be scaled up) for each construct

 i. This will be one biological replicate

6. 48 hr after transfection, remove medium from cells, wash with PBS, and lyse in 1 ml/well of mammalian protein extraction reagent containing protease inhibitor cocktail (aprotinin, AEBSF, leupeptin and pepstatin A, all at 1:1000) and phosphatase inhibitor cocktails (NaOV, Pepstatin A, Leupeptin, AEBSF, NaF, aprotinin) at 4°C or on ice.

7. Collect lysate and sonicate.

 a. Perform test for optimal conditions as follows.

 i. Sonication for 5 min

 ii. Sonication for 10 min

 b. Centrifuge lysate in refrigerated microcentrifuge at 14000x$g$ at 4°C for 10 min.

 c. Collect supernatants and measure the protein concentration of each using the DC Protein Assay Kit II according to the manufacturer's instructions.

 d. Will need >50 µg total protein to proceed

 i. If 50 µg total protein is not achieved the reaction will be scaled to a 25 cm plate. These conditions will be used for the subsequent replicates without any further optimization.

 ii. If further optimization is needed, the experiment will not proceed to step 7 until this is achieved.

 e. Aliquot lysate protein for measuring IDH oxidative (Step 9) and reductive activity (step 10) and for examining expression of $IDH2^{WT}$, $IDH2^{R172K}$ by western blot (step 11).

8. Measuring IDH oxidative activity:

 a. Mix 0.3 µg of each protein lysate with 200 µl of assay buffer solution in a 96-well plate. Each condition should be plated in triplicate.

 i. Assay buffer solution: 100 mM Tris-HCl buffer (pH 7.5), 1.3 mM $MnCl_2$, 0.33 mM EDTA, 0.1 mM ß-$NADP^+$, 0.1 mM D-(+)-*threo*-isocitrate

 ii. Include buffer lacking lysate protein to determine background reading.

 b. Put mixtures in spectrometer and measure absorbance at 340 nm every 20 s for 30 min.

 c. Use absorbance readings at 5 min intervals for analysis.

 i. An exploratory investigation of all data will be used in the analysis as well.

9. Measuring IDH reductive activity:

 a. Mix 3 µg of each protein lysate with 200 µl of assay buffer solution in a 96-well plate. Each condition should be plated in triplicate.

 i. Assay buffer solution: 100 mM Tris-HCl buffer (ph 7.5), 1.3 mM $MnCl_2$, 0.01 mM ß-NADPH, 0.5 mM $\alpha$-ketoglutarate

 ii. Include buffer lacking lysate protein to determine background reading.

 b. Put mixtures in spectrometer and measure absorbance at 340 nm every 20 min for 3 hr.

10. Western blot to confirm protein expression:

 a. Add sample buffer and boil lysates to prepare for loading.

 i. Sample buffer: 0.5 mL 1 M TrisCl, pH 6.8, 1 mL glycerol, 0.5 mL ß-mercaptoethanol, 0.24 g SDS, 0.1 mL 1% bromophenol blue.

 ii. Add 30 µg of protein per well by diluting protein to same concentrations (based on protein quantification results) in 10 µL of lyse buffer and added 20 µL of sample buffer

 iii. Place at 65°C for 15 min.

 b. Separate 20–30 µg of protein per lane on an 8% SDS-PAGE gel with protein ladder.

 i. Run through the stacker at 45 mAmp/gel, then increase to 300 V for 3 hr.

 c. Transfer to nitrocellulose membrane.

 i. Transfer at 100 A for 1 hr 40 min in 2.5 mM Tris, 19 mM glycine in 20% methanol.

 ii. Wash membrane in deionized water then wash in 1X TBST.

 iii. Confirm protein transfer with Ponceau stain.

 d. Block membrane with 5% milk/0.2% azide in TBST for 30 min at room temperature.

 e. Incubate with the following primary antibodies using the manufacturer's recommended dilution. Following antibodies will be probed at one time

 i. Mouse anti-IDH2; 37 kDa

 ii. Goat anti-IDH1; 47 kDa

 f. Incubate with appropriate secondary antibodies using manufacture's recommended dilutions

 i. HRP-conjugated sheep anti-mouse

 ii. HRP-conjugated rabbit anti-goat
 1. The anti-actin antibody is HRP conjugated and a secondary antibody incubation is not necessary.
 g. Treat membranes with ECL reagent according to manufacturer's recommendations and image.
 h. Between antibody incubations, inactivate HRP activity by incubating with a final concentration of 1mM sodium azide in blocking buffer.
 i. Shake at room temp for 1 hr.
 ii. Wash membrane 3 x 5 min in 1X TBST.
 iii. Incubate with ECL reagent as directed by the manufacturer and image at a time point of at least 5 min to confirm HRP inactivation
 iv. Save blank image
 i. Incubate with Rabbit anti-actin-HRP; 45 kDa [additional] to evaluate loading control
 j. Treat membranes with ECL reagent according to manufacturer's recommendations and image.
11. Repeat steps 6–9 independently six additional times.

## Deliverables

- Data to be collected:
  - Sequencing reads and agarose gel images confirming vector identity and integrity
    - pcDNA3
    - pcDNA3-IDH2$^{WT}$
    - pcDNA3-IDH2$^{R172K}$
  - Raw data from plate reader for reduced NADP$^+$ and oxidated NADPH
  - Background subtracted readings
  - Full western images, including ladder
    - Ponceau stains confirming protein transfer
    - ECL negative control from step 9-hr

## Confirmatory analysis plan

Statistical Analysis of the Replication Data:

- Note: At the time of analysis, we will perform the Shapiro-Wilk test and generate a quantile-quantile plot to assess the normality of the data. We will also perform Levene's test to assess homoscedasticity. If the data appears skewed, we will perform the appropriate transformation to proceed with the proposed statistical analysis. If this is not possible, we will perform the equivalent non-parametric Wilcoxon-Mann-Whitney test.
  - For oxidative activity assays:
    - Bonferroni corrected ANOVA followed by two-tailed Bonferroni corrected planned contrasts:
      - Vector vs. IDH2$^{WT}$
      - Vector vs. IDH2$^{R172K}$
  - For reductive activity assays
    - Bonferroni corrected ANOVA followed by two-tailed Bonferroni corrected planned contrasts:
      - Vector vs. IDH2$^{WT}$
      - Vector vs. IDH2$^{R172K}$
  - Western blot:
    - This is a quality control experiment and is not powered to detect a specific effect.
- Meta-analysis of original and replication attempt:
  - This replication attempt will perform the statistical analysis listed above, compute the effects sizes, compare them against the reported effect size in the original paper and use a meta-analytic approach to combine the original and replication effects, which will be presented as a forest plot.

## Known differences from the original study

Although not performed by the original authors, actin was added as internal loading control for Western blots and will be added to the resulting data. Details of the Western blot protocol and possible stripping/sodium azide treatment were unspecified; information was added by the replicating lab. The details of the transfection specifics were unspecified and that information is provided by the replicating lab. Additionally, these experiments will be conducted in 6-well dishes, however, if total protein yield is not sufficient, the replicating lab will scale up to 25 cm dishes.

## Provisions for quality control

All data obtained from the experiment - raw data, data analysis, control data, and quality control data - will be made publicly available, either in the published manuscript or as an open access data-set available on the Open Science Framework (https://osf.io/8l4ea/).

- STR profiling and mycoplasma testing results
- Sequencing reads and agarose gel images confirming vector identity and integrity
- Ponceau stains confirming protein transfer for Western Blot
- Confirmation of HRP inactivation prior to proceeding with the following antibodies.

## Protocol 2: Production of 2-HG from IDH2 WT and mutant transfected cells

In this protocol, the production of 2-HG from 293T cells transfected with vectors expressing IDH2$^{WT}$ or IDH2$^{R172K}$ is measured by gas chromatography-mass spectrometry (as seen in Figures 3A–C). The amount of 2-HG relative to glutamate is quantified, as seen in Figure 3D.

### Sampling

- Experiment will be performed with at least three biological replicates for a final power of at least 80%. The original data are qualitative, thus to determine an appropriate number of replicates to initially perform, sample sizes based on a range of potential variance was determined.
  - See Power Calculations section for details.
- Experiment has three conditions:
  - Condition 1: 293T cells expressing IDH2$^{WT}$
  - Condition 2: 293T cells expressing IDH2$^{R172K}$
  - Condition 3: 293T cells expressing empty pCDNA3 vector
- For each condition, lysates will be analyzed for 2-HG/glutamate levels

### Materials and reagents

| Reagent | Type | Manufacturer | Catalog # | Comments |
|---|---|---|---|---|
| 293T cells | Cells | ATCC | CRL-3216 | Original source unspecified |
| Dulbecco's modified Eagle's medium (DMEM) | Media | Invitrogen | 11965118 | Original unspecified |
| Pen/Strep | Reagent | Fisher | 15140-122 | Original unspecified |
| FBS | Reagent | Hyclone | SH30071.03 | Replaces FBS from CellGro |
| pcDNA-IDH2$^{WT}$ | Plasmid | Generated in Protocol 1 | | |
| pcDNA-IDH2$^{R172K}$ | Plasmid | Generated in Protocol 1 | | |
| Lipofectamine 2000 | Reagent | Invitrogen | 11668027 | |
| Methanol | Reagent | Fisher | A452SK-4 | Original unspecified |

*Continued on next page*

*Continued*

| Reagent | Type | Manufacturer | Catalog # | Comments |
|---|---|---|---|---|
| Refrigerated centrifuge | Equipment | Labnet International, Inc | PrismR | Original unspecified |
| Nitrogen gas | Reagent | Generated in lab | | Original unspecified |
| AG-1 X8 100-200 anion exchange column | Reagent | Bio-Rad | 731-6211 | Poly-Prep Columns, AG 1-X8, chloride form |
| HCl | Reagent | Fisher | SA56-1 | Original unspecified |
| N-methyl-N-tert-butyldimethylsily trifluoroacetamide (MTBSTFA; Regis) | Reagent | Regis | 1-270243-200 | |
| Gas Chromatograph with an HP-5MS capillary column and Mass selective detector | Equipment | Agilent 7890A with 7693 Autosampler | | |
| Cold trap concentrator | Equiptment | Labconco Centrivap | | |
| R(-)-2-HG | Reagent | Sigma-Aldrich | H8378-100MG | Original unspecified |

## Procedure

### Notes

- 293T cells grown in DMEM supplemented with 10% FBS at 37°C in 5% $CO_2$.
- All cells will be sent for STR profiling and mycoplasma testing

1. Seed 0.25–1 x $10^6$ 293T cells per well of a 6-well plate in growth medium without antibiotics.
   a. Grow overnight.
   b. Confirm cells at 70–80% confluence by light microscopy at time of transfection.
2. Transfect 293T cells with pCDNA3, pCDNA3-IDH2$^{WT}$, or pCDNA3-IDH2$^{R172K}$ with Lipofect-amine 2000 according to manufacturer instructions.
   a. Transfect 1 µg of plasmid DNA per well in 6-well plate at 70–80% confluence.
   b. Generate duplicate plates for each transfection:
      i.   Harvest one plate at 24 hr.
      ii.  Harvest one plate at 48 hr.
3. 24 hr later, replace with fresh media with 1x pen/strep
4. 24 or 48 hr later, gently remove medium from proliferating cells.
   a. Note: from this point on this protocol contains information as described in (*Bennett et al., 2008*).
5. Rapidly quench cells with 1–2 ml per well of -80°C methanol.
   a. Chill cells to -80°C and incubate at -80°C for 15 min.
6. Scrape cells off the dish and transfer the cell suspension to a 15 ml conical tube.
   a. Centrifuge for 5 min at 2000x*g* at 4°C to pellet cellular debris.
   b. Transfer supernatant to a fresh 15 ml tube.
7. Resuspend the pellet in 500 µl of -80°C 80% methanol in water by vortexing.
   a. Incubate at 4°C for 15 min.
   b. Centrifuge for 5 min at 2000x*g* at 4°C.
   c. Combine supernatant with supernatant from Step 6b.
   d. Repeat step 7 for a third round of extraction and combine all supernatants.
8. Evaporate to dryness using a cold trap concentrator.
9. Elute through an AG-1 X8 100–200 anion exchange resin according to the manufacturer's instructions.
   a. Wash with five column volumes of wash buffer.
   b. Elute in 3N HCl.
10. Evaporate to dryness using cold trap concentrator
11. Redissolve sample in MSTFA + FAME.
    a. Prepare 40 mg/mL Methoxyamine hydrochloride (MeOX) solution in pyridine.

 i. Weigh out methoxyamine hydrochloride in 1.5 ml Eppendorf tube on balance and add appropriate amount of pyridine.
- b. Vortex MeOX solution and sonicate at 60°C for 15 min to dissolve.
- c. Add 10 μl of 40 mg/ml MeOX solution to each dried sample.
- d. Shake at maximum speed at 60°C for 1 hr.
- e. To 1 ml of MSTFA, add 10 μl of FAME marker.
  - i. Vortex for 10 s.
- f. Add 91 μl of MSTFA + FAME mixture to each sample and standard. Cap immediately.
  - i. Shake at maximum speed at 37°C.
- g. Transfer contents to glass vials with micro-inserts and cap immediately.
  - i. Submit to GCTOF MS analysis.
12. Inject samples into GC-MS.
    - a. Operate the detector in spitless mode using electron impact ionization.
      - i. Ionizing voltage: -70 eV
      - ii. Electron multiplier: 1060 V
    - b. GC temperature ramp:
      - i. Hold at 100°C for 3 min.
      - ii. Ramp to 230°C at 4°C/min.
      - iii. Hold for 4 min.
      - iv. Ramp to 300°C.
      - v. Hold for 5 min.
    - c. Record mass range of 50–500 amu and record 2.71 scans/s.
13. Repeat steps 1–12 independently three additional times.

## Deliverables

- ▪ Data to be collected:
  - • 24 hr samples:
    - ○ GC traces for all samples run
      - • Close-up of the time range showing metabolite abundance for aspartate, glutamate, and 2-HG for cells transfected with IDH2$^{WT}$ (Figure 3A) and cells transfected with IDH2$^{R172K}$ (Figure 3B).
        - ○ Mass spectrum confirmation of metabolite identity as 2-HG.
  - • 48 hr run
    - ○ GC traces for all samples run
      Close-up of the time range showing metabolite abundance for aspartate, glutamate, and 2-HG for cells transfected with IDH2$^{WT}$ (Figure 3A) and cells transfected with IDH2$^{R172K}$ (Figure 3B).
    - ○ Quantification of the relative intensity of the 2-HG signal to the glutamate signal, graphed as seen in Figure 3D.

## Confirmatory analysis plan

- • Statistical Analysis of the Replication Data:
- • Note: At the time of analysis, we will perform the Shapiro-Wilk test and generate a quantile-quantile plot to assess the normality of the data. We will also perform Levene's test to assess homoscedasticity. If the data appears skewed, we will perform the appropriate transformation to proceed with the proposed statistical analysis. If this is not possible, we will perform the equivalent non-parametric test.
  - • Two-way ANOVA performed on 2-HG/glutamate ratios followed by Fisher's LSD for the following comparisons:

    - • Vector vs. IDH2$^{WT}$
    - • IDH2$^{WT}$ vs. IDH2$^{R172K}$

    o Analyses will be performed on both 24 and 48 hr runs.
- • Meta-analysis of original and replication attempt:
  - • The replication data will be presented as a mean with 95% confidence intervals and will include the original data point, calculated directly from the graph, as a single point on the same plot for comparison.

## Known differences from the original study

- The GC-MS sample preparation protocol was modified by the replicating lab including a shaking incubation step at 11f. However, this protocol was taken from Bennett et al. which the authors reference in the original manuscript.

## Provisions for quality control

All data obtained from the experiment - raw data, data analysis, control data and quality control data - will be made publicly available, either in the published manuscript or as an open access data-set available on the Open Science Framework (https://osf.io/8l4ea/).

- STR profiling and mycoplasma testing results.
- Mass spectrum of the metabolite peak for derivatized 2HG to confirm identity.

## Protocol 3: Assessing the correlation of IDH status with 2-HG levels in samples from patients with AML

In this protocol, samples from patients with acute myeloid leukemia (AML) are examined for their IDH mutational status and their level of 2-HG, as seen in Figure 5.

### Sampling

- This experiment will use four samples per group for a final power of at least 80%.
  - See Power Calculations section for details.
- This experiment has three genetically distinct groups:
  - AML patients with no *IDH* mutations
  - AML patients with mutant *IDH1*
  - AML patients with mutant *IDH2,* including both R172K and R140Q mutants
- All samples will come from Roswell Park Cancer Institute and are ficoll separated in media with 10% DMSO and prescreened for *IDH* genotypic status.
- Each patient sample will be assessed for their ratio of 2-HG/glutamate.

### Materials and reagents

| Reagent | Type | Manufacturer | Catalog # | Comments |
|---|---|---|---|---|
| Samples of peripheral blood, bone marrow, or pheresis from patients with karyotypically normal AML | Patient sample | NA | NA | Banked RPCI samples |
| DMSO | Reagent | Fisher | BP231-1 | Original Unspecified |
| Methanol | Reagent | Fisher | A452SK-4 | Original unspecified |
| Refrigerated centrifuge | Equipment | Labnet International, Inc | PrismR | Original unspecified |
| AG-1 X8 100-200 anion exchange column | Reagent | Bio-Rad | 731-6211 | Poly-Prep Columns, AG 1-X8, chloride form |
| HCl | Reagent | Fisher | SA56-1 | Original unspecified |
| N-methyl-N-tert-butyldimethylsily trifluoroacetamide (MTBSTFA; Regis) | Reagent | Regis | 1-270243-200 | |
| Gas Chromatograph with an HP-5MS capillary column and Mass selective detector | Equipment | Agilent 7890A with 7693 Autosampler | | |
| Cold trap concentrator | Equiptment | Labconco Centrivap | | |

## Procedure

1. GC-MS analysis of 2-HG levels.
   a. If using frozen cells, warm cells to 37°C in a 37°C water bath for 10 min
   b. Centrifuge cells for 5 min at 1000x*g* to form a pellet
      i. If necessary, transfer cells to a conical or microcentrifuge tube
   c. Gently remove freezing medium from MNCs
   d. Proceed with metabolite extraction and GC-MS analysis as detailed in protocol 2 Steps 5 through 12.
   e. For each sample, divide the GC signal intensity of their 2-HG peak by the signal intensity of their glutamate peak and graph.

## Deliverables

- Data to be collected:
  - Tabulated patient data (age, sex, *IDH* mutation status, 2-HG/glutamate ratio) (as seen in Table 1)
  - GC traces for all samples
  - Graph of 2-HG/glutamate ratio for samples by mutational status, as seen in Figure 5C.

## Confirmatory analysis plan

- Statistical Analysis of the Replication Data:
- Note: The authors report WT IDH ratios were less than 1% which we are using as the constant for the comparisons below.
  - Bonferroni Correct one-sample t-test for 3 comparisons (alpha corrected for 2 test groups = 0.025)
    - Constant vs. $IDH1^{mutant}$
    - Constant vs. $IDH2^{mutant}$
    - Constant vs $IDH1/2^{mutants}$
- Meta-analysis of original and replication attempt:
  - This replication attempt will perform the statistical analysis listed above, compute the effects sizes, compare them against the reported effect size in the original paper and use a meta-analytic approach to combine the original and replication effects, which will be presented as a forest plot.

## Known differences from the original study

- The GC-MS sample preparation protocol was modified by the replicating lab including a shaking incubation step at 11f, protocol 2. However, this protocol was taken from Bennett et al. which the authors reference in the original manuscript.

## Provisions for quality control

All data obtained from the experiment - raw data, data analysis, control data and quality control data - will be made publicly available, either in the published manuscript or as an open access dataset available on the Open Science Framework (https://osf.io/8l4ea/). This includes confirmation of the GCMS peaks and elution times as well as MS QC data.

## Power calculations

For details of power calculations, see spreadsheet and additional files at https://osf.io/9jkpg/

### Protocol 1
Summary of original data estimated from graph reported in Figure 2A:

- SD was calculated using formula SD = SEM*(SQRT n=3).

| Sample | Time | Mean | SEM | SD |
|---|---|---|---|---|
| IDH2$^{WT}$ | 0 | 0 | 0.0820 | 0.1421 |
| | 5 | 0.225 | 0.0820 | 0.1421 |
| | 10 | 0.45 | 0.1025 | 0.1776 |
| | 15 | 0.679 | 0.1538 | 0.2664 |
| | 20 | 0.917 | 0.1974 | 0.3419 |
| | 25 | 1.129 | 0.2512 | 0.4352 |
| | 30 | 1.342 | 0.3 | 0.5196 |
| IDH2$^{R172K}$ | 0 | 0 | 0.0820 | 0.1421 |
| | 5 | 0.038 | 0.0820 | 0.1421 |
| | 10 | 0.062 | 0.0820 | 0.1421 |
| | 15 | 0.062 | 0.0820 | 0.1421 |
| | 20 | 0.062 | 0.0820 | 0.1421 |
| | 25 | 0.1 | 0.0820 | 0.1421 |
| | 30 | 0.096 | 0.0820 | 0.1421 |
| Vector | 0 | 0 | 0.0564 | 0.0977 |
| | 5 | 0.021 | 0.0564 | 0.0977 |
| | 10 | 0.021 | 0.0564 | 0.0977 |
| | 15 | 0.017 | 0.0564 | 0.0977 |
| | 20 | 0.017 | 0.0564 | 0.0977 |
| | 25 | 0.033 | 0.0564 | 0.0977 |
| | 30 | 0.021 | 0.0564 | 0.0977 |

Linear regression to determine slopes from estimate values.
Calculations performed with R software (version 3.2.2) (*R Core Team, 2015*)

| Sample | Mean slope | SD | N |
|---|---|---|---|
| IDH2$^{WT}$ | 0.01 | 0.090 | 3 |
| IDH2$^{R172K}$ | 0.06 | 0.140 | 3 |
| Vector | 0.67 | 0.280 | 3 |

Summary of original data estimated from graph reported in Figure 2B:

- SD was calculated using formula SD = SEM*(SQRT(n)), where n = 3.

| Sample | Time | Original_Value_Mean | SEM | SD |
|---|---|---|---|---|
| IDH2$^{WT}$ | 0 | 0 | 0.0039 | 0.0067 |
| | 17 | -0.003 | 0.0060 | 0.0105 |
| | 33 | -0.004 | 0.0073 | 0.0126 |
| | 50 | -0.005 | 0.0102 | 0.0177 |
| | 71 | -0.006 | 0.0104 | 0.0181 |
| | 90 | -0.008 | 0.0114 | 0.0198 |
| | 112 | -0.009 | 0.0117 | 0.0202 |
| | 131 | -0.01 | 0.0075 | 0.0130 |
| | 171 | -0.014 | 0.0121 | 0.0211 |

*Continued on next page*

*Continued*

| Sample | Time | Original_Value_Mean | SEM | SD |
|---|---|---|---|---|
| IDH2$^{R172K}$ | 0 | 0 | 0.0039 | 0.0067 |
| | 17 | -0.006 | 0.0039 | 0.0067 |
| | 33 | -0.009 | 0.0065 | 0.0114 |
| | 50 | -0.016 | 0.0085 | 0.0147 |
| | 71 | -0.024 | 0.0080 | 0.0139 |
| | 90 | -0.028 | 0.0087 | 0.0152 |
| | 112 | -0.036 | 0.0095 | 0.0164 |
| | 131 | -0.043 | 0.0104 | 0.0181 |
| | 171 | -0.055 | 0.0095 | 0.0164 |
| Vector | 0 | 0 | 0.0026 | 0.0046 |
| | 17 | 0.001 | 0.0026 | 0.0046 |
| | 33 | 0 | 0.0026 | 0.0046 |
| | 50 | 0 | 0.0026 | 0.0046 |
| | 71 | 0 | 0.0026 | 0.0046 |
| | 90 | 0 | 0.0026 | 0.0046 |
| | 112 | -0.002 | 0.0026 | 0.0046 |
| | 131 | -0.002 | 0.0026 | 0.0046 |
| | 171 | -0.003 | 0.0026 | 0.0046 |

Linear regression to determine slopes from estimates values.
Calculations performed with R software (version 3.2.2) (*R Core Team, 2015*)

| Sample | Mean slope | SD | N |
|---|---|---|---|
| IDH2$^{WT}$ | -0.0006 | 0.005 | 3 |
| IDH2$^{R172K}$ | -0.0241 | 0.013 | 3 |
| Vector | -0.0065 | 0.016 | 3 |

## Test family

- One-way ANOVA: Fixed effects, omnibus, one-way: Bonferroni correction: alpha error = 0.025.

## Power calculations

- Power calculations were performed using G*Power, version 3.1.7 (*Faul et al., 2007*).
- ANOVA F test statistic and partial $\eta^2$ performed with R software, version 3.2.2 (*R Core Team, 2015*).

| Groups | F test statistic | Partial $\eta^2$ | Effect size *f* | A priori power | Total sample size |
|---|---|---|---|---|---|
| Slopes of NADPH production from IDH2$^{WT}$, IDH2$^{R172}$, or Vector (Figure 2A) | F(2,6) = 10.8 | 0.7826 | 1.897636 | 99.99%[1] | 21[1] (3 groups) |
| Slopes of NADP$^+$ production from IDH2$^{WT}$, IDH2$^{R172}$, or Vector (Figure 2B) | F(2,6) = 3.02 | 0.5023 | 1.0048 | 94.13%[1] | 21[1] (3 groups) |

[1] 7 samples per group will be used based on the planned comparisons making the power at least 80%.

## Test family

- ▪ 2 tailed *t* test, Wilcoxon-Mann-Whitney test, Bonferroni's correction: alpha error = 0.0125

Power Calculations performed with G*Power software, version 3.1.7 (*Faul et al., 2007*).

### Figure 2A (NADPH production) values

| Group 1 | Group 2 | Effect size *d* | A priori power | Group 1 sample size | Group 2 sample size |
|---|---|---|---|---|---|
| Vector | IDH2^WT | 3.05134 | 98.8%[1] | 7[1] | 7[1] |
| Vector | IDH2^R172K | 2.12463[2] | 80.0%[2] | 7 | 7 |

[1] 7 samples per group will be used based on the Vector vs IDH2^R172K NADP^+ planned comparison making the power 98.8%.

[2] A sensitivity calculation was performed since the original data showed a non-significant effect. This is the effect size that can be detected with 80% power and the indicated sample size. The original effect size reported was 0.49386.

### Figure 2B (NADP^+ production) values

| Group 1 | Group 2 | Effect size *d* | A priori power | Group 1 sample size | Group 2 sample size |
|---|---|---|---|---|---|
| Vector | IDH2^WT | 2.12463[1] | 80.0%[1] | 7 | 7 |
| Vector | IDH2^R172K | 2.21471 | 89.3% | 7 | 7 |

[1] A sensitivity calculation was performed since the original data showed a non-significant effect. This is the effect size that can be detected with 80% power and the indicated sample size. The original effect size reported was 0.47369.

## Test family

- • Due to the large variance, these parametric tests are only used for comparison purposes. To ensure an adequate sample size is used, the number is based on the non-parametric tests listed above.
- • 2 tailed *t* test, difference between two independent means, Bonferroni's correction: alpha error = 0.0125

Power Calculations performed with G*Power software, version 3.1.7 (*Faul et al., 2007*).

### Figure 2A (NADPH production) values

| Group 1 | Group 2 | Effect size *d* | A priori power | Group 1 sample size | Group 2 sample size |
|---|---|---|---|---|---|
| Vector | IDH2^WT | 3.05134 | 99.2%[1] | 7[1] | 7[1] |
| Vector | IDH2^R172K | 2.03[2] | 80.0%[2] | 7 | 7 |

[1] Seven samples per group will be used based on the Vector vs IDH2^R172K NADP^+ planned comparison making the power 98.8%.

[2] A sensitivity calculation was performed since the original data showed a non-significant effect. This is the effect size that can be detected with 80% power and the indicated sample size. The original effect size reported was 0.33972.

### Figure 2B (NADP^+ production) values

| Group 1 | Group 2 | Effect size *d* | A priori power | Group 1 sample size | Group 2 sample size |
|---|---|---|---|---|---|
| Vector | IDH2^WT | 2.05829[1] | 80.0%[1] | 7 | 7 |
| Vector | IDH2^R172K | 2.03 | 90.4% | 7 | 7 |

[1] A sensitivity calculation was performed since the original data showed a non-significant effect. This is the effect size that can be detected with 80% power and the indicated sample size. The original effect size reported was 0.51213.

## Protocol 2: Figure 3D

Summary of original data

- Note: data estimated from published graphs

| Sample | Mean intracellular 2-HG/glutamate | Assumed N |
|---|---|---|
| Vector | 0.0105 | 3 |
| IDH2$^{WT}$ | 0.0102 | 3 |
| IDH2$^{R172K}$ | 1.2 | 3 |

### Test family

- One way ANOVA followed by Bonferroni corrected planed comparisons:
  - Power calculations:
    - Vector vs. IDH2$^{R172K}$
    - IDH2$^{WT}$ vs. IDH2$^{R172K}$
  - Sensitivity Calculations
    - Vector vs. IDH2$^{WT}$

### Power calculations

- Power calculations were performed using GraphPad PRISM v6 and G*Power (version 3.1.7) (*Faul et al., 2007*)
- Because the data did not display variance, we have performed power calculations with a range of variances and an assumed N of 3 per group.
- 2% variance

**ANOVA; α=0.05**

| F(2,6) | Partial eta2 | Effect size f | Power | Total N |
|---|---|---|---|---|
| 7370 | 0.999593 | 49.55807 | >99.99% | 6* |

Power calculations; α=0.05

| Group 1 | Group 2 | Effect size d | Power | N/group |
|---|---|---|---|---|
| Vector | IDH2$^{WT}$ | 70.10710478 | >99.99% | 2* |
| IDH2$^{WT}$ | IDH2$^{R172K}$ | 70.08927663 | >99.99% | 2* |

Sensitivity Calculations; α=0.05, powered to 80%

| Group 1 | Group 2 | Effect size d | Detectable d | N/group |
|---|---|---|---|---|
| Vector | IDH2$^{R172K}$ | 1.449123183 | 0.2774844 | 3 |

*With a minimum of 3 per group (9 total), achieved power is >99.99%.

- 15% variance

**ANOVA; α=0.05**

| F(2,6) | Partial eta2 | Effect size f | Power | Total N |
|---|---|---|---|---|
| 131 | 0.977612 | 6.608085 | 99.99% | 6* |

Power calculations; α=0.05

| Group 1 | Group 2 | Effect size d | Power | N/group |
|---|---|---|---|---|

*Continued on next page*

*Continued*

**ANOVA; α=0.05**

| Vector | IDH2<sup>WT</sup> | 9.347613971 | 98.65% | 2* |
|---|---|---|---|---|
| IDH2<sup>WT</sup> | IDH2<sup>R172K</sup> | 9.345236884 | 98.65% | 2* |

Sensitivity Calculations; α=0.05, powered to 80%

| Group 1 | Group 2 | Effect size d | Detectable d | N/group |
|---|---|---|---|---|
| Vector | IDH2<sup>R172K</sup> | 0.193216424 | 0.0539826 | 3 |

*With a minimum of 3 per group (9 total), achieved power is >99.99%.

- 28% variance

**ANOVA; α=0.05**

| F(2,6) | Partial eta2 | Effect size f | Power | Total N |
|---|---|---|---|---|
| 37.60 | 0.926108 | 3.540235 | 98.61% | 6* |

Power calculations; α=0.05

| Group 1 | Group 2 | Effect size d | Power | N/group |
|---|---|---|---|---|
| Vector | IDH2<sup>WT</sup> | 5.007650342 | 99.28% | 3 |
| IDH2<sup>WT</sup> | IDH2<sup>R172K</sup> | 5.006376902 | 99.28% | 3 |

Sensitivity Calculations; α=0.05, powered to 80%

| Group 1 | Group 2 | Effect size d | Detectable d | N/group |
|---|---|---|---|---|
| Vector | IDH2<sup>R172K</sup> | 0.103508799 | 0.0511419 | 3 |

*With a minimum of 3 per group (9 total), achieved power is 99.99%.

- 40% variance

**ANOVA; α=0.05**

| F(2,6) | Partial eta2 | Effect size f | Power | Total N |
|---|---|---|---|---|
| 18.43 | 0.860009 | 2.478571 | 85.73% | 6* |

Power calculations; α=0.05

| Group 1 | Group 2 | Effect size d | Power | N/group |
|---|---|---|---|---|
| Vector | IDH2<sup>WT</sup> | 3.505355239 | 88.73% | 3 |
| IDH2<sup>WT</sup> | IDH2R172K | 4.205285771 | 96.37% | 3 |

Sensitivity Calculations; α=0.05, powered to 80%

| Group 1 | Group 2 | Effect size d | Detectable d | N/group |
|---|---|---|---|---|
| Vector | IDH2<sup>R172K</sup> | 0.072456159 | 0.0505594 | 3 |

*With a minimum of 3 per group (9 total), achieved power is 99.92%.

- In order to produce quantitative replication data, we will run the experiment three times. We will determine the standard deviation across the biological replicates and combine this with the reported value from the original study to simulate the original effect size. We will use this simulated effect size to determine the number of replicates necessary to reach a power of at least 80%. We will then perform additional replicates, if required, to ensure that the experiment has more than 80% power to detect the original effect.
- Note: Simulation analysis was also conducted using randomly generated values based on the SD and variance desired. These data are comparable to what is seen above when using a parametric model approach. Also there may be a need to appropriately transform these data based on the scale of Figure 3D, and we have assumed that this is one representative sample and not averages of all the data showing no variance. This simulation will be loaded to the OSF (https://osf.io/8l4ea/).

## Protocol 3: Figure 5C

### Summary of original data

- Note: data estimated from published graphs and log transformed. Data includes IDH$^{WT}$ (no mutations in *IDH1* or *IDH2*), IDH1$^{R132C/G}$, IDH2$^{Mutant}$ (IDH2$^{R172K}$ and IDH2$^{R140Q}$)

| Sample | 2HG/glutamate | log(2HG/glut) |
|---|---|---|
| IDH$^{WT}$(Constant) | 0.01 | -4.605 |
| IDH1$^{Mutant}$ | 0.600 | -0.511 |
| IDH1$^{Mutant}$ | 1.200 | 0.182 |
| IDH1$^{Mutant}$ | 1.600 | 0.470 |
| IDH1$^{Mutant}$ | 1.800 | 0.588 |
| IDH1$^{Mutant}$ | 3.000 | 1.099 |
| IDH1$^{Mutant}$ | 0.600 | -0.511 |
| IDH2$^{Mutant}$ | 0.140 | -1.966 |
| IDH2$^{Mutant}$ | 0.160 | -1.832 |
| IDH2$^{Mutant}$ | 0.290 | -1.237 |
| IDH2$^{Mutant}$ | 0.300 | -1.204 |
| IDH2$^{Mutant}$ | 0.310 | -1.171 |
| IDH2$^{Mutant}$ | 0.470 | -0.755 |
| IDH2$^{Mutant}$ | 0.590 | -0.528 |
| IDH2$^{Mutant}$ | 0.310 | -1.171 |

### Test family

- One sample t-test comparing Constant and mutant *IDH* groups:
  - Constant vs. *IDH1*$^{R132C/G}$
  - Constant vs. *IDH2*$^{mutant}$ (grouped)
  - Constant vs. *IDH1/2*$^{mutant}$ (grouped)

### Power calculations

Power calculations were performed using R software version 3.2.2 and G*Power (version 3.1.7) (***Faul et al., 2007***). Bonferroni corrected one-sample *t*-tests compared to. 01 (threshed as reported by original authors).

| Constant | Group | Effect size d | A priori power | Group sample size |
|---|---|---|---|---|
| 0.01 | IDH1$^{R132C/G}$ | 8.404 | 99.99% | 4 |
| 0.01 | IDH2$^{Mutant}$ | 6.746 | 99.99% | 4 |
| 0.01 | IDH1/2$^{Mutant}$ | 4.361 | 99.99% | 4 |

- Because of the inherent complications that can occur when using primary patient cell lines, we have adjusted our sample size to four samples/group even though we achieve >90% power when using three samples/group.

## Acknowledgements

The Reproducibility Project: Cancer Biology core team would like to thank Courtney Soderberg at the Center for Open Science for assistance with statistical analyses. We would also like to thank

Kermit L. Carraway III, Kacey Vandervorst, and Jason Hatakeyama from the Department of Biochemistry and Molecular Medicine at UC Davis and the UC Davis Comprehensive Cancer Center for methods consultation. The following companies generously donated reagents to the Reproducibility Project: Cancer Biology; American Type and Tissue Collection (ATCC), Applied Biological Materials, BioLegend, Charles River Laboratories, Corning Incorporated, DDC Medical, EMD Millipore, Harlan Laboratories, LI-COR Biosciences, Mirus Bio, Novus Biologicals, Sigma-Aldrich, and System Biosciences (SBI).

## Additional information

### Group author details

Reproducibility Project: Cancer Biology

Elizabeth Iorns: Science Exchange, Palo Alto, United States; William Gunn: Mendeley, London, United Kingdom; Fraser Tan: Science Exchange, Palo Alto, United States; Joelle Lomax: Science Exchange, Palo Alto, United States; Stephen Williams: Center for Open Science, Charlottesville, Virginia; Nicole Perfito: Science Exchange, Palo Alto, United States; Timothy Errington: Center for Open Science, Charlottesville, Virginia

### Competing interests
OF, MRS: West Coast Metabolomics Center is a Science Exchange associated laboratory RP:CB: EI, FT, JL, NP: Employed by and hold shares in Science Exchange Inc. The other authors declare that no competing interests exist.

### Funding

The Reproducibility Project: Cancer Biology is funded by the Laura and John Arnold Foundation, provided to the Center for Open Science in collaboration with Science Exchange. The funder had no role in study design or the decision to submit the work for publication.

### Author contributions
OF, MRS, CES-T, Drafting or revising the article; RP:CB, Conception and design, Drafting or revising the article

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
