## [Decision Letter]

Thank you for submitting your work entitled "Registered report: The common feature of leukemia-associated IDH1 and IDH2 mutations is a neomorphic enzyme activity converting α-ketoglutarate to 2-hydroxyglutarate" for consideration by *eLife*. Your article has been reviewed by three peer reviewers, and the evaluation has been overseen by Jessica Tyler as Reviewing Editor and Vivek Malhotra as the Senior Editor.

The reviewers have discussed the reviews with one another and the Reviewing Editor has drafted this decision to help you perform your study, and then submit your revised submission once the experiments are complete.

Summary:

The submitted report plan will replicate the results of the key experiments in Ward et al., 2010, using three protocols to replicate the experiments. Power calculations and confirmatory analysis plans are described. In general the authors have done a great job, and the proposed analyses are appropriate and accurate, but we have the following suggestions:

Essential revisions:

1) In the confirmatory analysis plan of the Protocols 2 and 3, One-way ANOVA is followed by planned comparisons using Fisher's LSD and in the corresponding power calcualtions for the t-tests of Protocol 2, a 0.05 α error is used. Fisher's LSD however does not control the family wise error rate (see Hayter, 1986) and it is useful only for the calculation of the effect size d. If that is the same statistical analysis method that was used in the original paper, we would like you to do it, but we would also like you to perform a Bonferroni correction (α = 0.025), as this is superior.

Anthony J. Hayter. The maximum familywise error rate of fisher's least significant difference test. Journal of the American Statistical Association, 81(396):

1000-1004, 1986. doi: 10.1080/01621459.1986.10478364.

2) Rather than combining the *IDH1* mutant and *IDH2* mutant patient samples into a single group (table after “Power calculations” in subheading “Protocol 3: Figure 5C”), it will be useful to also consider them as two separate groups in addition to *IDH1*+*IDH2*, as the difference in subcellular localization may lead to differences in D2HG abundance, and it would be worth knowing if there are differences in *IDH1* mutants vs. *IDH2* mutants, in addition to control comparisons. This would obviously change the statistical comparison needed to analyze the data.

For your interest, we wanted to communicate to you some insightful suggestions that the reviewers had to improve the impact of the findings of the analyses beyond the original study. We appreciate that that is not the intent of the reproduction goal of the Registered report, but we include those ideas here as they may be worthy of your consideration for increasing the impact of this study and/or your future work:

1) It may be useful to include *IDH2 ^R140Q^* mutants in the cell based studies, as it is likely that some AML patient samples will harbor that specific mutation.

2) An estimation of non-*IDH1,2*-formed 2-HG levels would be useful.

3) Distinction between the R and S chiral forms of 2-HG would be useful.

4) Concerning the Protocol 1, i.e. replication of the Figure 2 experiments from the original article, this test represents biochemical assays of somewhat "reconstituted" WT or mutant *IDH2* forms and is of the appropriate design. The design could be improved only by a few variations of the overexpressed *IDH2*, either WT or mutant, to better correlate the possible variations due to the different levels of expression between WT or mutant. The statistics to be performed is sufficient to encompass the quality of the data and results even if the overexpression strength is not varied.

5) Concerning the Protocol 2, i.e. replication of the Figure 3 experiments there is a problem whether "mere reproducibility" is tested or whether a scientific design of this experiment is to be scrutinized. In the former case, assessment is similar as for the Protocol 1. In the latter case however, the experimental design cannot exclude the possibility that WT enzyme does not form 2-HG, since conditions for this counter-Krebs cycle direction of reaction are not set up in the experimental design. Actually such conditions were set up if 2-HG was assayed in experiments similar to those in protocol 1, but long-term - where both directions of reaction are assayed in parallel. The Protocol 2 also cannot rule out the possibility, that mutant enzyme occurring under conditions leading to the forward-Krebs cycle direction is as inefficient in synthesizing 2-HG as the WT.

6) Concerning the Protocol 3, i.e. replication of the Figure 5 experiments analyzing the AML patient samples the precise error-free identification of mutations is the key to good reproducibility. Since this effort should rather mimic a clinical study, you should do the best to maximize the quality of these replication analyses and should substantially increase the number of samples per group. The suggested 4 samples per group are fulfilling a power of 80%, which however only represents a single repeat of the figure from the original publication but not validation of the possible diagnostic testing.

7) It would potentially be more beneficial to also focus on other aspects related to D or L2HG and *IDHs* such as the effects on epigenetics and the therapeutic potential of targeting 2HG-mediated epigenetic alterations.

---

## [Author Response]

*1) In the confirmatory analysis plan of the Protocols 2 and 3, One-way ANOVA is followed by planned comparisons using Fisher's LSD and in the corresponding power calcualtions for the t-tests of Protocol 2, a 0.05 α error is used. Fisher's LSD however does not control the family wise error rate (see Hayter, 1986) and it is useful only for the calculation of the effect size d. If that is the same statistical analysis method that was used in the original paper, we would like you to do it, but we would also like you to perform a Bonferroni correction (α = 0.025), as this is superior.*

*Anthony J. Hayter. The maximum familywise error rate of fisher's least significant difference test. Journal of the American Statistical Association, 81(396): 1000-1004, 1986. doi: 10.1080/01621459.1986.10478364.*

With regard to Protocol 2, we agree with the reviewers’ comment on the use of a correction, such as Bonferroni or the modification of LSD by Hayter to control for the MFWER, however as Hayter describes in his 1986 paper, this applies in situations where the ANOVA is unbalanced or with a balanced design with four or more populations. Since the proposed analysis is balanced with three population groups, the LSD is sufficiently conservative and powerful to account for the multiple comparisons in this specific situation. This is further explained by Levin et al., 1994 and discussed in Maxwell and Delaney, 200 (Chapter 5) and Cohen, 2001 (Chapter 12).

References:

Levin, J.R., Serline, R.C., & Seaman M.A. (1994). A controlled, powerful multiple-comparison strategy for several situations. Psychological Bulletin, 115, 153-159.

Maxwell, S.E. & Delaney, H.D. (2004). Designing experiments and analyzing data: a model comparison perspecitive. Lawrence Erlbaum Associates, Mahwah, N.J., 2nd edition.

Cohen, B.H. (2001). Explaining psychological statistics. John Wiley and Sons, New York, 2nd edition.

As for Protocol 3, we have clarified this section. We will perform 3 one sample t-tests compared to the constant (a threshold of 0.01(WT) as defined by Ward et al.). The comparisons will be constant vs. *IDH1* mutant, constant vs. *IDH2* mutant, and constant vs. *IDH1/2* pooled mutants.

2) Rather than combining the IDH1 mutant and IDH2 mutant patient samples into a single group (table after “Power calculations” in subheading “Protocol 3: Figure 5C”), it will be useful to also consider them as two separate groups in addition to IDH1+IDH2, as the difference in subcellular localization may lead to differences in D2HG abundance, and it would be worth knowing if there are differences in IDH1 mutants vs. IDH2 mutants, in addition to control comparisons. This would obviously change the statistical comparison needed to analyze the data.

We apologize for the confusion here. There was a typo in the table. Only *IDH2* mutants will be combined for all analyses because of the likely low percentage of samples that will harbor the *IDH2^R172^* allele. Further we plan on combining all IDH mutation and compare them to WT. With regard to the *IDH1* vs. *IDH2* mutants, because the Ward et al. paper never made these comparisons this analysis would be outside of the scope of the proposal. However, all data from this project will be made publically available to allow for further exploratory analysis such as this.

For your interest, we wanted to communicate to you some insightful suggestions that the reviewers had to improve the impact of the findings of the analyses beyond the original study. We appreciate that that is not the intent of the reproduction goal of the Registered report, but we include those ideas here as they may be worthy of your consideration for increasing the impact of this study and/or your future work:

*1) It may be useful to include IDH2 ^R140Q^ mutants in the cell based studies, as it is likely that some AML patient samples will harbor that specific mutation.*

We apologize if this was not made clear initially. Any *IDH2 ^R140Q^* mutants that are identified in our sampling will be included in the replication of original Figure 5C. An addition has been made to the Sampling section of Protocol 3 to clarify this.

*2) An estimation of non-IDH1,2-formed 2-HG levels would be useful.*

We agree with the reviewers that this would be an interesting investigation. While outside of the scope of this specific project, all data from GC-MS will be made publically available to allow for further exploratory analysis.

*3) Distinction between the R and S chiral forms of 2-HG would be useful.*

We agree with the reviewers that this would be an interesting investigation. While outside of the scope of this specific project, all data from GC-MS will be made publically available to allow for further exploratory analysis.

*4) Concerning the Protocol 1, i.e. replication of the Figure 2 experiments from the original article, this test represents a biochemical assays of somewhat "reconstituted" WT or mutant IDH2 forms and is of the appropriate design. The design could be improved only by a few variations of the overexpressed IDH2, either WT or mutant, to better correlate the possible variations due to the different levels of expression between WT or mutant. The statistics to be performed is sufficient to encompass the quality of the data and results even if the overexpression strength is not varied.*

We agree that investigating the dose dependent impact of *IDH* overexpression on NADP-NADPH and NADPH-NADP would be interesting, however it is outside of the scope of this project as the original authors did not perform this experiment.

*5) Concerning the Protocol 2, i.e. replication of the Figure 3 experiments there is a problem whether "mere reproducibility" is tested or whether a scientific design of this experiment is to be scrutinized. In the former case, assessment is similar as for the Protocol 1. In the latter case however, the experimental design cannot exclude the possibility that WT enzyme does not form 2-HG, since conditions for this counter-Krebs cycle direction of reaction are not set up in the experimental design. Actually such conditions were set up if 2-HG was assayed in experiments similar to those in protocol 1, but long-term - where both directions of reaction are assayed in parallel. The Protocol 2 also cannot rule out the possibility, that mutant enzyme occurring under conditions leading to the forward-Krebs cycle direction is as inefficient in synthesizing 2-HG as the WT.*

We agree with the reviewers that this could be the case and we have done our best to design the experiments detailed in this protocol to be a “direct” replication. Study design is always an aspect to be investigated but like the other analyses, all data will be made public so that future analyses can take into account these aspects.

*6) Concerning the Protocol 3, i.e. replication of the Figure 5 experiments analyzing the AML patient samples the precise error-free identification of mutations is the key to good reproducibility. Since this effort should rather mimic a clinical study, you should do the best to maximize the quality of these replication analyses and should substantially increase the number of samples per group. The suggested 4 samples per group are fulfilling a power of 80%, which however only represents a single repeat of the figure from the original publication but not validation of the possible diagnostic testing.*

We understand the reviewers’ concerns about identification of mutational status and believe this has essential clinical implications. Here, our replication attempt is meant specifically to address the difference in 2HG/glutamate in *IDH* mutants vs. controls as reported in the original paper and not necessarily to investigate the precision of using 2HG/glutamate to identify *IDH* mutations, which we agree would require a larger sample size. The cell lines that we are obtaining are Roswell Park Cancer Institute and mutations have been confirmed multiple times on different platforms. Based on our power analyses, 4 samples should achieve 99% power to detect the published differences in 2HG/glutamate between WT and mutant *IDH*, as the effect size is quite large. Please keep in mind that the scale of Figure 5C is logarithmic.

7) It would potentially be more beneficial to also focus on other aspects related to D or L2HG and IDHs such as the effects on epigenetics and the therapeutic potential of targeting 2HG-mediated epigenetic alterations.

We agree that these are interesting avenues of investigation, however, they are outside of the scope of this project.